# Cognitive Function in Patients with Psychotic and Affective Disorders: Effects of Combining Pharmacotherapy with Cognitive Remediation

**DOI:** 10.3390/jcm13164843

**Published:** 2024-08-16

**Authors:** Eva I. J. Maihofer, Gabriele Sachs, Andreas Erfurth

**Affiliations:** 1Medical University of Vienna, 1090 Vienna, Austria; eva.maihofer@gesundheitsverbund.at (E.I.J.M.); gabriele.sachs@meduniwien.ac.at (G.S.); 21st Department of Psychiatry and Psychotherapeutic Medicine, Klinik Hietzing, 1130 Vienna, Austria

**Keywords:** schizophrenia, bipolar disorder, depression, cognitive screening, cognitive remediation, acute care psychiatry, cross-diagnostic assessment

## Abstract

**Background**: Cognitive impairment is a relevant problem in psychiatry and can be well assessed with a cross-diagnostic test such as the Screen for Cognitive Impairment in Psychiatry (SCIP). The aim of our pilot study is to assess cognitive impairment in acute psychiatric inpatients diagnosed with psychotic disorders, bipolar disorder and depression using the German version of the SCIP (SCIP-G). We also investigate whether cognitive dysfunction improves over the course of the inpatient treatment, where patients are offered a combination of pharmacological treatment and cognitive remediation. **Methods**: A total of 143 adult inpatients were included in the study. Cognitive testing was performed using two different forms of the SCIP-G. All patients received state-of-the-art pharmacotherapy and cognitive remediation using the COGPACK^®^ software package version 6.06. **Results**: Based on the ICD-10 Criteria for Research, 54 patients were given an F2 diagnosis (schizophrenia and schizotypal and delusional disorders). Thirty-nine patients met the criteria for bipolar disorder (F30 and F31) and fifty for depression (F32 and F33). At baseline, a significant difference was observed between the SCIP total scores of the F2 and F32/33 patients (*p* < 0.001) and between the F2 and F30/31 groups (*p* = 0.022). At the second measurement time point, the SCIP total score showed significant improvement in all three groups (*p* < 0.001), and there was no statistically significant interaction between SCIP total score and diagnostic groups (*p* = 0.860). **Conclusions**: Cognitive dysfunction is present in psychiatric disorders and can be easily assessed during an inpatient hospital stay. In our sample, patients with a psychotic disorder were more cognitively impaired at baseline than patients with an affective disorder. Inpatient treatment, consisting of pharmacotherapy and cognitive remediation, improved cognitive deficits. Patients with psychotic disorders, bipolar disorder and depression showed similar improvements in cognitive performance.

## 1. Introduction

Cognitive dysfunction is well established as a key feature in patients meeting diagnostic criteria for major psychiatric disorders, including schizophrenia [1,2,3,4,5,6], bipolar disorder [2,7,8,9] and depression [10,11,12,13]. While on the one hand a cross-diagnostic approach to symptoms and syndromes seems increasingly useful [14,15,16,17], on the other hand a precise analysis of potential differences between patients diagnosed with psychiatric disorders, both in the acute situation and in the course of the disorder, is necessary [2,18], especially with regard to functional outcome [9,19,20,21,22,23].

A variety of instruments have been found to be useful in assessing cognitive impairment in psychiatric disorders [24]. Some test batteries provide differentiated information about the whole range of cognitive and social cognitive domains, e.g., the MATRICS Consensus Cognitive Battery [25]. Other instruments have their strength in the rapid detection of symptoms of a specific disorder in the sense of a screening instrument (e.g., Brief Assessment of Cognition in Schizophrenia [26,27], and for transnosographic studies, those scales are particularly helpful that have been constructed independently of differentiating diagnostic features, such as the Screen for Cognitive Impairment in Psychiatry (SCIP) [28,29,30,31].

To improve cognition in patients, particularly those with schizophrenia, it has been proposed to combine medication with neuropsychological training [32,33,34,35,36,37]. The beneficial effects of cognitive remediation programmes in schizophrenia have been demonstrated [38], including effects on neuronal activation using functional neuroimaging studies [39,40]. The beneficial effects of cognitive remediation have also been described in patients with depressive disorders [41,42] and bipolar disorders [43,44].

The aim of our pilot study is to assess cognitive impairment in acute psychiatric inpatients diagnosed with psychotic disorders, bipolar disorder and depression using a cross-diagnostic screen such as the SCIP in its German version (SCIP-G) [30,45]. In a further step, we investigate whether cognitive dysfunction can be improved in the course of acute inpatient treatment by providing a fixed combination of pharmacological treatment with cognitive remediation through COGPACK^®^ (Marker software, Heidelberg & Ladenburg, Germany) [46].

## 2. Methods

### 2.1. Subjects

The 1st Department of Psychiatry and Psychotherapeutic Medicine, Klinik Hietzing, Vienna (formerly the 6th Department of Psychiatry, Otto-Wagner-Spital, Vienna, Austria) is one of 8 regional psychiatric departments in Vienna and is responsible for the emergency initial assessment and subsequent inpatient psychiatric care of adults from the Viennese municipal districts 12, 13 and 23 (currently about 280,000 inhabitants). Between 34 and 60 beds were available for this purpose during the study period. The sample consists of routinely collected clinical data of adult psychiatric patients residing in this specific catchment area and previously indicated for acute admission. A total of 143 adult inpatients with psychotic disorders (ICD-10: F2 diagnosis of schizophrenia and schizotypal and delusional disorders), bipolar disorder (ICD-10: F30/F31), or depression (ICD-10: F32/F33) were enrolled. All patients were fluent in German. Patients older than 65 years were excluded due to the increased likelihood of age-related mild cognitive impairment [47]. Other exclusion criteria were (a) a verbal IQ of less than 80 points as estimated using the Mehrfachwahl-Wortschatz-Intelligenztest (MWT-B) score [48], (b) a concomitant diagnosis of current substance abuse, or (c) aggressive behaviour. Patients who did not complete the screening and follow-up for any reason were not included in the analysis.

The study was conducted in accordance with the ethical principles of the Declaration of Helsinki and Good Clinical Practice. The study protocol was approved by the Ethics Committee of the City of Vienna.

### 2.2. Assessments

Diagnostic classification was carried out by specialists in psychiatry and psychotherapeutic medicine according to the Criteria for Research of the ICD-10 [49]; all other assessments were made by clinical psychologists.

Premorbid intelligence was estimated using the Mehrfachwahl-Wortschatz-Intelligenztest (MWT-B) [48]. The neuropsychological assessment was carried out at two points in time. To ensure comparability of data and to allow patients to adjust to the inpatient stay, the first test was administered between day 7 and day 14 of the stay. The timing of the second test was set as close as possible to discharge, not least to minimise learning effects [30].

The cognitive tests were administered using two different forms of the SCIP-G [30]. The Screen for Cognitive Impairment in Psychiatry, which is available in 3 equivalent forms for the purpose of repeated testing, is “designed for rapid and objective quantification of cognitive impairment” [28]. The test consists of 5 subscales assessing verbal learning, working memory, verbal fluency, verbal learning (delayed) and processing speed.

### 2.3. Cognitive Remediation Training

All patients received state-of-the-art pharmacotherapy plus cognitive remediation using a standardised computer-based training programme delivered using the COGPACK^®^ software package version 6.06 [46]. The COGPACK^®^ programme includes neurocognitive exercises that can be used to train vigilance, memory, perception, reaction, and visuomotor skills, as well as linguistic, intellectual, every-day, educational and professional performance.

### 2.4. Statistical Analysis

Data were analysed using the Statistical Package of Social Sciences (SPSS Inc. 27th version, Chicago, IL, USA).

To determine differences in sociodemographic, clinical and neuropsychological functioning between the three groups, one-way analysis of variance (ANOVA) was carried out for the continuous variables and chi-square analysis for the categorical variables. Post hoc tests were performed using Tukey’s post hoc test.

To investigate whether patients with different diagnoses differed in their cognitive performance across the two time points, a mixed factorial ANOVA was conducted.

## 3. Results

### 3.1. Demographic and Clinical Characteristics

A total of 143 patients were included in the study. According to the ICD-10 Criteria for Research, 54 patients were diagnosed with schizophrenia or schizotypal or delusional disorder (F2). Thirty-nine patients met the criteria for bipolar disorder (F30 and F31) and fifty for depression (F32 and F33). Table 1 shows the diagnostic sub-characterisation within the three main categories mentioned.

A total of 84 (58.7%) patients were female and 59 were male. Age ranged from 18 to 64 years, with a mean age of 37.91 years (SD = 12.69). The mean number of years of education was 14.27 (SD = 3.51), ranging from 7 to 24 years. The estimation of the premorbid intelligence level yielded a mean value of 27.86 (SD = 5.43) in the MWT-B (ranging from 15 to 36). Table 2 shows the distribution of the characteristics in the three diagnostic groups.

We performed a one-way ANOVA to assess group differences between psychotic, bipolar and depressed patients. There were no outliers, as shown by checking with a box plot. Equal variances were assumed, and homogeneity of variances was tested with Levene’s test (*p* < 0.05). The patient groups did not differ significantly in age (F = 2.050, df = 2, *p* = 0.133, η^2^ = 0.028), years of education (F = 1.023, df = 2, *p* = 0.362, η^2^ = 0.014) or MWT-B score (F = 1.563, df = 2, *p* = 0.213, η^2^ = 0.022). The patient groups also showed no difference in gender (chi-squared test: *p* = 0.705).

### 3.2. Neurocognitive Performance

SCIP total scores at baseline

Table 3 (time point 1) shows the SCIP total scores in the three diagnostic groups. As described above, a one-way ANOVA was performed to assess group differences between psychotic, bipolar, and depressed patients. SCIP total scores differed significantly between the three diagnostic groups (F (2,140) = 7.119, *p* < 0.001, η^2^ = 0.092). Tukey post hoc analysis revealed a significant difference between the SCIP total scores of psychotic and depressed patients (*p* < 0.001) and between the psychotic and bipolar groups (*p* = 0.022). No significant group difference was found between bipolar and depressive diagnoses (*p* = 0.801).

Within the bipolar group, no statistically significant difference was found when comparing the SCIP total scores of manic (F30.2/F31.0/F31.1/F31.2/F31.8, *n* = 22) and depressive (F31.3/F31.4/F31.5, *n* = 15) patients (SCIP manic episode: mean 70.77, SD = 12.313; SCIP depressive episode: mean 72.00, SD = 9.658, *p* = 0.748).

SCIP subscale scores at baseline

Figure 1 shows the SCIP subscale scores at baseline comparing psychotic, bipolar and depressed patients. Table 4 compares the subscale scores within the three diagnostic groups. A significant difference between the diagnostic groups is found in the “verbal learning” subtest (ANOVA: F = 5.215, df = 2, *p* = 0.007), as well as in the domains “working memory” (F = 4.047, df = 2, *p* = 0.020), “verbal fluency” (F = 3.33, df = 2, *p* = 0.39) and “verbal learning—delayed” (F = 4.797, df = 2, *p* = 0.010), but not in “processing speed” (F = 2.72, df = 2, *p* = 0.069).

Time interval between the two assessments

The mean time interval between the two SCIP assessments was 50.23 days (SD = 35.36). The three diagnostic groups did not differ significantly in their time interval (F = 0.625, df = 2, *p* = 0.537, η^2^ = 0.009).

Comparison of the SCIP total values at the two measurement time points

Figure 2 and Table 3 show the effect of combined treatment on SCIP total scores in the three diagnostic groups. The SCIP total score improved significantly in all three groups. In the mixed factorial ANOVA, there was a significant main effect for the SCIP total score (Greenhouse–Geisser F = 121.207, df = 1, *p* < 0.001, partial η^2^ = 0.464). There was no difference between diagnostic groups in terms of improvement: there was no statistically significant interaction between SCIP total score and diagnostic group (Greenhouse–Geisser F = 0.150, df = 2, *p* = 0.860, partial η^2^ = 0.002).

## 4. Discussion

Our data show that in all patient groups (psychotic disorders, bipolar disorder and depression) there is a significant increase in SCIP total scores after state-of-the-art pharmacotherapy plus cognitive remediation. This improvement in cognitive function did not differ between diagnostic groups. To our knowledge, this is the first study to demonstrate improvement in cognitive function after combined treatment with psychotropic medication and cognitive remediation in the three main diagnostic categories of psychotic disorders, bipolar disorder and depression in non-selected patients from acute care psychiatry.

Our study suggests that cognitive deficits are measurable in this population, and that the SCIP is well suited as a brief screening test to detect deficits and to identify subsequent remediation needs. As previously described [28,50,51], the SCIP allows a comparison across different psychiatric diagnostic groups; our data confirm that patients with psychotic disorders show greater cognitive impairment than affective patients [2]. The baseline SCIP total score in the psychotic disorder group of 64.98 (SD = 12.872) is very similar to that of schizophrenic and schizoaffective patients in the German validation study [30], where two forms of the SCIP yielded a mean total score of 62.57 (SD = 9.51) for form 1 and 66.53 (SD = 11.81) for form 2.

While the phenomenon that patients with psychotic disorders may perform worse in terms of neurocognition than those with affective disorders is well established [2,52,53], future studies will examine the role of negative symptoms [54,55,56] in mediating this effect [57] and the extent to which they influence the prospects for cognitive improvement through cognitive remediation.

SCIP allows a before-and-after comparison [30], which is useful for quality assurance in psychiatric care [58]. Indeed, many patients, especially those with affective disorders, but also individual psychotic patients, achieved test scores in the second test that would be expected of healthy subjects; in the German validation study [30], total scores of 87.57 (SD 6.71) and 92.60 (SD 7.02) were found in healthy subjects for two forms of the SCIP. As no randomised trial was conducted, it is not possible to determine whether this improvement represents a specific effect of cognitive remediation or an effect of the general treatment including appropriate pharmacotherapy.

Discussing the results of a cognitive test with the individual patient can increase the willingness to deal with one’s own illness and take responsibility for one’s own course, also with the additional help of psychoeducation [9]. The degree of cognitive improvement we observed in this study is similar across diagnostic groups, so it could be concluded that the offer of cognitive remediation should be made equally to all diagnostic groups, as it is equally promising for all groups.

Our patients represent the group of critically ill patients from their catchment area who are acutely admitted [59]. They were selected only if they could be motivated to cooperate and showed adherence to treatment [60]. Aggressive and agitated patients could not participate in this study. Agitation thus remains a particular challenge in the treatment of psychiatric patients [61,62], also with regard to the possible lack of recognition of their cognitive deficits [63,64] and the consequent lack of implementation of cognitive remediation programmes.

Limitations

Our pilot study describes improvements in cognitive performance over time; randomisation was not carried out. Future randomised controlled trials in acute psychiatry could investigate the specific effect of cognitive rehabilitation interventions as an adjunct to usual care in different diagnostic groups, using cross-diagnostic tests such as the SCIP.

## 5. Conclusions

Cognitive dysfunction is present in psychiatric disorders and can be easily assessed during inpatient hospitalisation.

Inpatient treatment consisting of pharmacotherapy and cognitive remediation improves cognitive deficits in non-selected, most severely ill psychiatric patients. Patients with psychotic disorders, bipolar disorder and depression show similar improvements in cognitive performance.

## Figures and Tables

**Figure 1 jcm-13-04843-f001:**
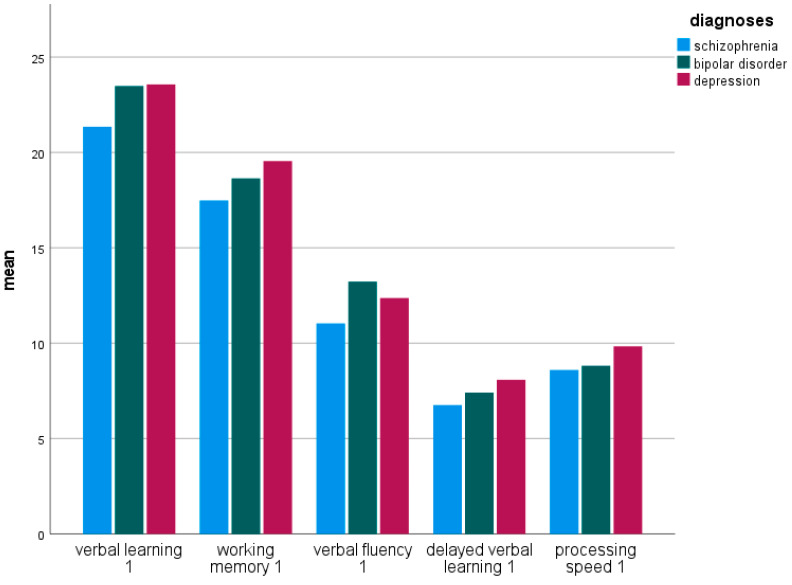
Neurocognitive performance of psychiatric patients at time point 1 (baseline): SCIP subscales in the comparison of the three diagnostic groups F2 (schizophrenia, schizotypal and delusional disorders, *n* = 54), F30/31 (bipolar disorder, *n* = 39) and F32/33 (depression, *n* = 50).

**Figure 2 jcm-13-04843-f002:**
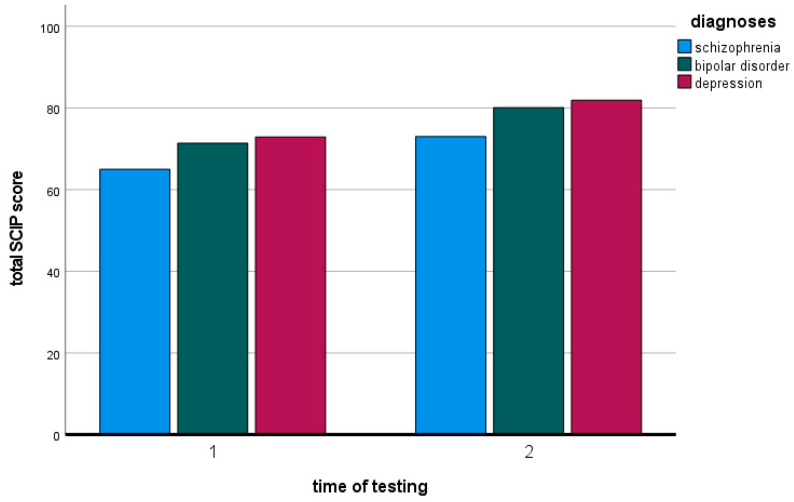
Effect of combined treatment on means of SCIP totals in the three diagnostic groups F2 (schizophrenia, schizotypal and delusional disorders), F30/31 (bipolar disorder) and F32/33 (depression).

**Table 1 jcm-13-04843-t001:** Diagnostic distribution of patients according to the ICD-10 Criteria for Research (World Health Organization 2011).

ICD-10 Code	Diagnosis	*n*	%
F20.0	Paranoid schizophrenia	30	55.6
F20.1	Hebephrenic schizophrenia	2	3.7
F20.3	Undifferentiated schizophrenia	1	1.9
F22.0	Delusional disorder	1	1.9
F23.0	Acute polymorphic psychotic disorder without symptoms of schizophrenia	1	1.9
F23.1	Acute polymorphic psychotic disorder with symptoms of schizophrenia	4	7.4
F23.8	Other acute and transient psychotic disorders	1	1.9
F25.0	Schizoaffective disorder, manic type	3	5.6
F25.1	Schizoaffective disorder, depressive type	7	13.0
F25.2	Schizoaffective disorder, mixed type	4	7.4
F2 total	Schizophrenia, schizotypal and delusional disorders	54	100.0
F30.2	Mania with psychotic symptoms	3	7.7
F31.0	Bipolar affective disorder, current episode hypomanic	4	10.3
F31.1	Bipolar affective disorder, current episode manic without psychotic symptoms	2	5.1
F31.2	Bipolar affective disorder, current episode manic with psychotic symptoms	12	30.8
F31.8	Other bipolar affective disorders (Recurrent manic episodes)	1	2.6
Mania subtotal		22	56.4
F31.3	Bipolar affective disorder, current episode mild or moderate depression	6	15.4
F31.4	Bipolar affective disorder, current episode severe depression without psychotic symptoms	7	17.9
F31.5	Bipolar affective disorder, current episode severe depression with psychotic symptoms	2	5.1
Bipolar depressed subtotal		15	38.5
F31.6	Bipolar affective disorder, current episode mixed	2	5.1
F30/31 total	Manic episode/Bipolar affective disorder	39	100.0
F32.1	Moderate depressive episode	6	12.0
F32.2	Severe depressive episode without psychotic symptoms	11	22.0
F32.3	Severe depressive episode with psychotic symptoms	5	10.0
F33.1	Recurrent depressive disorder, current episode moderate	6	12.0
F33.2	Recurrent depressive disorder, current episode severe without psychotic symptoms	20	40.0
F33.3	Recurrent depressive disorder, current episode severe with psychotic symptoms	2	4.0
F32/33 total	Depressive episode/Recurrent depressive disorder	50	100.0

**Table 2 jcm-13-04843-t002:** Distribution of gender, age, premorbid intelligence (as measured by the MWT-B) and years of education in the three diagnostic groups F2 (psychotic disorders), F30/31 (bipolar disorder) and F32/33 (depression). MWT-B: Mehrfachwahl-Wortschatz-Intelligenztest [48].

Diagnosis	Total *n*	Female *n*	Male *n*	Age Mean (SD)	MWT-B Mean (SD)	Years of Education Mean (SD)
psychotic disorders	54	30	24	35.41 (10.93)	27.11 (5.67)	14.26 (3.51)
bipolar disorder	39	25	14	40.56 (14.56)	28.85 (4.49)	14.99 (3.65)
depression	50	29	21	38.70 (12.45)	28.50 (4.99)	13.94 (3.29)

**Table 3 jcm-13-04843-t003:** SCIP totals of patients in the three diagnostic groups F2 (psychotic disorders), F30/31 (bipolar disorder) and F32/33 (depression) before and after combination treatment.

		Timepoint 1			Timepoint 2			
Diagnosis	*n*	Mean Value	Standard Deviation	Range	Mean Value	Standard Deviation	Range	Improvement of Mean Values between Timepoints
psychotic disorders	54	64.98	12.872	36–89	73.00	11.808	42–100	8.02
bipolar disorder	39	71.38	10.946	52–95	80.10	11.805	58–107	8.72
depression	50	72.92	9.710	45–90	81.90	11.379	61–109	8.98

**Table 4 jcm-13-04843-t004:** Comparison of the scores of the subscales in the three diagnostic groups F2 (psychotic disorders, *n* = 54), F30/31 (bipolar disorder, *n* = 39) and F32/33 (depression, *n* = 50) at baseline.

Diagnosis	Verbal Learning Mean (SD)	Working Memory Mean (SD)	Verbal Fluency Mean (SD)	Verbal Learning—Delayed Mean (SD)	Processing Speed Mean (SD)
psychotic disorders	21.35 (4.344) *^#^	17.48 (4.437) ^#^	11.04 (4.273) *	6.76 (2.363) ^#^	8.59 (2.716) ^#^
bipolar disorder	23.49 (4.019) *	18.64 (3.166)	13.23 (4.457) *	7.41 (2.035)	8.82 (2.827)
depression	23.57 (3.266) ^#^	19.55 (3.163) ^#^	12.37 (3.712)	8.08 (2.029) ^#^	9.84 (2.946) ^#^

*: *p* < 0.05 in the comparison of F2 and F30/31; ^#^: *p* < 0.05 in the comparison of F2 and F32/33.

## Data Availability

The datasets generated for this study are available on request from the corresponding author.

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
