# Peer review of "Cognitive Function in Patients with Psychotic and Affective Disorders: Effects of Combining Pharmacotherapy with Cognitive Remediation"

_jcm, 2024, doi:10.3390/jcm13164843_

Round 1

Reviewer 1 Report

Comments and Suggestions for Authors

Introduction: it would be helpful to mention the beneficial effects of cognitive remediation in bipolar disorder and depression (with reference). The author only mention the effect in SZ. 

table 1. it is important to include percentages. 

in line 134 you use 3 decimals, it`s better only 2. 

Table 2: In first column I suggest change ICD code by diagnosis (example F32/33 by depressive disorders). this will be helpful to the readers. And I think the same in the manuscript. you can mention codes and diagnosis at first and then use the diagnosis. 

Table 3: the same suggestion as in table 2. Look up that you use the diagnoses in figure 1 and it looks better than the ICD code in tables. So, if you change the codes for diagnosis in results it would be better.

In table 3 seems that you compare mean values but you doesn`t include the p values nor statistical test. did you use parametric or non parametric analysis in this?. you must clarify that in results (lines 184-190).

table 4: same suggestion as in other tables. 

Comments on the Quality of English Language

you must improve redaction in results and discussion. 

Author Response

Introduction: it would be helpful to mention the beneficial effects of cognitive remediation in bipolar disorder and depression (with reference). The author only mention the effect in SZ. 

We thank the reviewer for bringing this to our attention. We have added the following sentence to the text: "The beneficial effects of cognitive remediation have also been described in patients with depressive disorders (Motter et al. 2016, Mokhtari et al. 2023) and bipolar disorders (Tsapekos et al. 2020, Tsapekos et al. 2023)."

Motter JN, Pimontel MA, Rindskopf D, Devanand DP, Doraiswamy PM, Sneed JR. Computerized cognitive training and functional recovery in major depressive disorder: A meta-analysis. J Affect Disord. 2016 Jan 1;189:184-91. doi: 10.1016/j.jad.2015.09.022. 

Mokhtari S, Mokhtari A, Bakizadeh F, Moradi A, Shalbafan M. Cognitive rehabilitation for improving cognitive functions and reducing the severity of depressive symptoms in adult patients with Major Depressive Disorder: a systematic review and meta-analysis of randomized controlled clinical trials. BMC Psychiatry. 2023 Jan 27;23(1):77. doi: 10.1186/s12888-023-04554-w.  

Tsapekos D, Seccomandi B, Mantingh T, Cella M, Wykes T, Young AH. Cognitive enhancement interventions for people with bipolar disorder: A systematic review of methodological quality, treatment approaches, and outcomes. Bipolar Disord. 2020 May;22(3):216-230. doi: 10.1111/bdi.12848.  

Tsapekos D, Strawbridge R, Cella M, Young AH, Wykes T. Does cognitive improvement translate into functional changes? Exploring the transfer mechanisms of cognitive remediation therapy for euthymic people with bipolar disorder. Psychol Med. 2023 Feb;53(3):936-944. doi: 10.1017/S0033291721002336.     

table 1. it is important to include percentages.   

Thank you for pointing this out. Table 1 has been supplemented with percentages.  

in line 134 you use 3 decimals, it`s better only 2. 

We have made the requested correction.

Table 2: In first column I suggest change ICD code by diagnosis (example F32/33 by depressive disorders). this will be helpful to the readers. And I think the same in the manuscript. you can mention codes and diagnosis at first and then use the diagnosis. 

We would like to thank you for your comment. We made the requested changes to table 2 and to the text (129-131).

Table 3: the same suggestion as in table 2. Look up that you use the diagnoses in figure 1 and it looks better than the ICD code in tables. So, if you change the codes for diagnosis in results it would be better.

We have implemented the requested changes. 

In table 3 seems that you compare mean values but you doesn`t include the p values nor statistical test. did you use parametric or non parametric analysis in this?. you must clarify that in results (lines 184-190).

To make the table clearer, we have included the statistical information in the text. As stated in 155-160, we performed a one-way ANOVA to assess group differences between F2, F30/31 and F32/F33 patients. SCIP total scores differed significantly between the three diagnostic groups (F(2,140)=7.119, p < .001, η² = .092). Tukey post-hoc analysis revealed a significant difference between the SCIP total scores of F2 and F32/33 patients (p< .001) and between the F2 and F30/31 groups (p=.022).

table 4: same suggestion as in other tables. 

Thank you again for this suggestion, we have made the requested change.

you must improve redaction in results and discussion. 

We have carefully edited the results and discussion sections on the basis of mother tongue (British English) and made appropriate changes to the manuscript.

Reviewer 2 Report

Comments and Suggestions for Authors

I think that this is a well written article on a subject of interest in psychiatry. 

I have a few questions for clarification.

1. The authors write that subjects were given optimal pharmacotherapy and cognitive remediation via an algorithmic system. However, they do not state whether the pharmacotherapy was directed specifically towards treating the primary psychiatric symptoms, or whether pharmacological treatment for the cognitive symptoms was included or was a primary focus.

2. There are a number of cognitive enhancers available, which I utilize in my practice as a geriatric psychiatrist. If none of these were utilized, I wonder why not? I have used them with success in patients for whom cognitive changes were secondary to a primary illness, and treatment of the primary illness was insufficient. Were they interested only in the use of psychological remediation methods?

3. How do they account for the similarity of improvement across the three treatment groups? Are they assuming that the same mechanism underlies the cognitive dysfunction is all three groups? Is there evidence to support such a claim? If not, and the mechanisms are different, then why is the level of improvement the same? Do they suggest that that is a effect of their psychological remediation method? If so, then it seems to me that they would need a control group who received only pharmacotherapy (if not directed at cognition) or a control group receiving non-cognition directed pharmacotherapy to compare against. In the absence of controls I don't see how they can claim that specific interventions made the difference.

Author Response

The authors write that subjects were given optimal pharmacotherapy and cognitive remediation via an algorithmic system. However, they do not state whether the pharmacotherapy was directed specifically towards treating the primary psychiatric symptoms, or whether pharmacological treatment for the cognitive symptoms was included or was a primary focus.

Thank you for bringing this aspect to our attention. The patients received the pharmacotherapy that the responsible senior physicians considered to be adequate in the acute therapy, based on an overall view of the symptoms and the expected range of adverse drug reactions. In the case of psychotic or manic patients, only atypical antipsychotics were used. Cognitive aspects were not included in the decision on peracute therapy, partly because in this situation it is usually not possible to assess cognition in a differentiated manner. On the other hand, the data currently available do not provide conclusive evidence that significant differences can be found between the atypicals with regard to cognition, so that all atypicals are equally recommended for schizophrenia in this regard (Vita et al. Eur Psychiatry. (2022) 65(1):e57.).    

There are a number of cognitive enhancers available, which I utilize in my practice as a geriatric psychiatrist. If none of these were utilized, I wonder why not? I have used them with success in patients for whom cognitive changes were secondary to a primary illness, and treatment of the primary illness was insufficient. Were they interested only in the use of psychological remediation methods?  

Thank you for pointing that out. The patients screened in our study had primarily acute pathologies that necessitated admission. The indication for cognitive remediation was part of the routine therapeutic efforts and not an expression of a cognitive dysfunction previously perceived by the treating physicians. With regard to cognitive enhancers: in Austria, the acetylcholinesterase inhibitors donepezil, rivastigmine, galantamine and the NMDA antagonist memantine are approved for the treatment of dementia and are reimbursed accordingly by the health insurance funds. They are not used outside of this indication in psychiatric acute care departments. Other putative cognitive enhancers such as Gingko biloba or piracetam are also not reimbursed.    

How do they account for the similarity of improvement across the three treatment groups? Are they assuming that the same mechanism underlies the cognitive dysfunction is all three groups? Is there evidence to support such a claim? If not, and the mechanisms are different, then why is the level of improvement the same? Do they suggest that that is a effect of their psychological remediation method? If so, then it seems to me that they would need a control group who received only pharmacotherapy (if not directed at cognition) or a control group receiving non-cognition directed pharmacotherapy to compare against. In the absence of controls I don't see how they can claim that specific interventions made the difference.  

Thank you for this important and interesting question. The genetic, molecular, biochemical and neurophysiological bases of cognition are diverse (for genetics, see e.g. Payton A. Investigating cognitive genetics and its implications for the treatment of cognitive deficit. Genes Brain Behav. 2006;5 Suppl 1:44-53. doi: 10.1111/j.1601-183X.2006.00194.x.) and, in the sense of personalised medicine, it can be assumed that specific elements may be associated with the presence and severity of a cognitive dysfunction in each individual. The clinically relevant first step is to determine whether a cognitive dysfunction is present, which we have done transdiagnostically with the SCIP. The second clinically interesting question is whether the existing and routinely offered resource COGPACK training should be reserved for patients in a particular diagnostic group or whether it should be offered to all patients in these three groups equally in clinical practice. In our opinion, our study has provided a preliminary answer to this question. Since the improvements were similar in all three groups, we want to offer COGPACK therapy to all patients in the three groups in our clinical practice in the future. 
Of course, we cannot answer the question of whether COGPACK had a positive (or negative) effect on its own, or whether the improvement in SCIP is an expression of the general clinical improvement (due to pharmacological effects, the time factor, the setting of the inpatient psychiatric admission ward, or other factors). 
We very much hope that the sentence "As no randomised trial was conducted, it is not possible to determine whether this improvement represents a specific effect of cognitive remediation or an effect of the general treatment including appropriate pharmacotherapymakes it clear that we do not claim that specific interventions made the difference.

Round 2

Reviewer 2 Report

Comments and Suggestions for Authors

Acceptable